# Expanding the Use of SGLT2 Inhibitors in T2D Patients Across Clinical Settings

**DOI:** 10.3390/cells14090668

**Published:** 2025-05-02

**Authors:** Alessandro Cuttone, Vittorio Cannavò, Raouf Mastan Sheik Abdullah, Pierluigi Fugazzotto, Giada Arena, Simona Brancati, Andrea Muscarà, Carmela Morace, Cristina Quartarone, Domenica Ruggeri, Giovanni Squadrito, Giuseppina Tiziana Russo

**Affiliations:** 1Department of Clinical and Experimental Medicine, University of Messina, 98100 Messina, Italy; vittorio.cannavo@hotmail.it (V.C.); abdulraouf.mastansheikabdullah@unime.it (R.M.S.A.); pierlu_96@hotmail.it (P.F.); giadarena96.ga@gmail.com (G.A.); simona.brancati@live.it (S.B.); andreamuscaraam@gmail.com (A.M.); carmela.morace@unime.it (C.M.); giovanni.squadrito@unime.it (G.S.); giuseppina.russo@unime.it (G.T.R.); 2Internal Medicine and Diabetology Unit, University Hospital of Messina, 98124 Messina, Italy; cristina.quartarone@polime.it (C.Q.); domenica.ruggeri@unime.it (D.R.)

**Keywords:** SGLT2i, T2D, CVOTs, cardiovascular effects, renal effects, bone metabolism, MASLD, uric acid, blood pressure, cognitive impairment

## Abstract

Sodium-glucose cotransporter-2 inhibitors (SGLT2i) are currently recommended in patients with type 2 diabetes (T2D) to reduce serum glucose levels. Moreover, robust evidence has clearly demonstrated their beneficial cardiovascular and renal effects, making this class of drugs pivotal for the treatment of T2D, especially when complicated by diabetic kidney disease or heart failure. However, several other comorbidities are frequently encountered in T2D patients beyond these long-term diabetes complications, especially in the internal medicine setting. For some of these comorbidities, such as MAFLD and cognitive impairment, the association with diabetes is increasingly recognized, with the hypothesis of a common pathophysiologic background, whereas, for others, a coincident epidemiology linked to the ageing of populations, including that of T2D subjects, may be advocated. In the effort of personalizing T2D treatment, evidence on the potential effects of SGLT2i in these different clinical conditions is accumulating. The purpose of this narrative review is to update current literature on the effects of SGLT2i for the treatment of T2D in different clinical settings beyond glycaemic control, and to elucidate potential molecular mechanisms by which they exert these effects.

## 1. Introduction

Type 2 diabetes mellitus (T2D) is a chronic and complex metabolic disease that represents a major public health problem, affecting approximately 10% of the population [1]. This disease is characterized by chronic hyperglycaemia because of insufficient insulin signalling due to insulin resistance associated with different degrees of defective insulin secretion, coupled with defects in endogenous glucose production (EGP), glucagon secretion, incretin effect, and renal handling of glucose excretion [2].

SGLT2i are FDA-approved for managing adult patients with T2D to improve blood sugar control adjunct to diet and exercise. 

Current international guidelines as the ADA guidelines 2025, recommend SGLT2i for people with T2D and established ASCVD or indicators of high ASCVD risk, HF, or CKD, independently of A1C values and metformin use, and in consideration of person-specific factors.

SGLT2i inhibit sodium-dependent glucose transporters-2 (SGLT2) that are located in the early proximal renal tubule, which are responsible for the reabsorption of most (80–90%) of the glucose filtered by the glomerulus [3]. The resulting increase in urinary glucose excretion lowers plasma glucose concentrations. This mechanism of action depends on blood glucose levels and is independent of the action and availability of insulin. This class of drugs reduces HbA1c by approximately 0.6–1.0% [4]. SGLT2i have a good tolerability, with common and uncommon adverse events including genitourinary tract infection, diabetic ketoacidosis (DKA), or hypotension [4].

Several molecules are today available within the class of SGLT2i, with a recently amplified indication of use, beyond glycaemic control. Empagliflozin has been approved by the FDA for T2D to reduce cardiovascular death in adults with T2D, treat adults with HF with reduced ejection fraction (HFrEF), and treat adults with HF and preserved ejection fraction. Canagliflozin is approved for T2D and DKD, and reduces the risk of heart attack, cardiovascular death, and stroke in adults with T2D and established cardiovascular disease, and reduces the risk of hospitalization for heart failure (HHF) in T2D and DKD patients. Dapagliflozin is indicated for T2D, to reduce the risk of HHF, for the treatment of HF in patients with HFrEF, and for CKD patients with or without the risk of T2D progression. Ertugliflozin is approved for T2D [5,6,7,8].

In the real-world setting, patients with T2D have several comorbidities with clinical features that cover different areas of internal medicine beyond cardiology and nephrology, ageing being a major contributor to this complexity.

In this narrative review, we provide an update on extra-glycaemic effects and clinical advantages of SGLT2i and their potential mechanism of action in different clinical settings commonly encountered in patients with T2D.

## 2. Effects of SGLT2i on Cardiovascular Outcomes

SGLT2i have an established impact on the management of T2D, heart failure (HF), and chronic kidney disease (CKD). Initially developed to improve glycaemic control, these pharmacological agents have demonstrated significant cardioprotective benefits across multiple populations, both diabetic and non-diabetic [9,10].

The beneficial effect that these molecules would have at the cardiac level could depend on several factors. Still suggestive and interesting is the hypothesis that these drugs may directly inhibit SGLT2, which is also present in the myocardium: the expression of the genes coding for this cotransporter has not yet been proven, and conflicting experiences exist in the literature.

In the study by Marfella et al., which evaluated the expression of SGLT2 in human cells from diabetic and non-diabetic patients and in an AC16 myocardiocyte line from patients undergoing heart transplantation using immunohistochemistry, immunofluorescence and SGLT2 quantization with both real-time reverse transcription-polymerase chain reaction and Western blot analysis showed the presence of SGLT2 in patients with end-stage decompensation and that this expression is overexpressed in patients with diabetes mellitus. Additionally, when cardiomyocytes were submitted to high concentrations of glucose, this led to overexpression of SGLT2 in these cells [11].

This overexpression was also demonstrated in the study by Scisciola et al. In this case, overexpression of the SGLT2 gene and protein was marked in patients with low flow, low gradient aortic stenosis (LF-LG AS), and this overexpression appeared to be related to important changes in cardiac metabolism: in particular, there was less utilization of fatty acids and their oxidation to produce energy, with greater utilization of glucose as an energy substrate and subsequent accumulation of lipids in the myocardium. This, in turn, led to increased expression of SGLT2, which led to a greater metabolic shift towards glucose utilization with reduced ATP production and functional failure of the myocardium itself. This naturally led to a progressive deterioration of cardiac function and failure [12].

These results were finally confirmed in a study on cardiomyocytes derived from human induced pluripotent stem cells (hiPSCs) that were given high doses of glucose to induce hypertrophy and were subsequently treated with empagliflozin: during the first phase of the study, there was increased expression of SGLT1, SGLT2 and NPPB, which also augmented with increasing cell size in immunofluorescence images, and this led to alterations in intracellular calcium concentrations that resulted in reduced sensitivity of myocardial fibres to this ion. Starting empagliflozin therapy showed a reduction in cytosolic calcium levels and a normalization of cardiomyocyte contractility, supporting the thesis of a direct effect of this molecule on the myocardiocyte independent of other metabolic activities [13].

In contrast, in a study that isolated atrial and ventricular myocardiocytes from 88 patients undergoing cardiac surgery and when these cells were submitted to an insulin-containing solution by assessing mRNA levels, no expression of SGLT2 was demonstrated, but only of SGLT1, GLUT1, and GLUT4 [14].

This has led several scholars to believe that the effect of these molecules is not due to their direct action on specific receptors present on the myocardium but to other side effects such as natriuresis/diuresis, improved cardiac energy metabolism due to the production of ketone bodies, inhibition of the sympathetic nervous system, inhibition of the Na/H-exchanger, improvement of hyperuricaemia, inhibition of SGLT1 decreasing epicardial fat mass, increasing erythropoietin levels, increasing circulating pro-vascular progenitor cells, decreasing oxidative stress, and improving vascular function [15].

### 2.1. SGLT2i and Acute and Chronic Heart Failure

Multiple trials (e.g., DAPA-HF, EMPEROR-Reduced, DECLARE-TIMI 58) have shown SGLT2 inhibitors’ effects in HFrEF, reducing HF hospitalizations by 30–40% and cardiovascular death by 20% [16,17,18].

Data from DECLARE-TIMI 58 concluded that dapagliflozin reduced cardiovascular death, HF hospitalization, and all-cause mortality specifically in HFrEF [16].

Banerjee [18] and Vaduganathan [19] showed reductions in HF hospitalization and CV outcomes in patients with ejection fractions ≥40%. Both dapagliflozin and empagliflozin significantly improved symptoms, quality of life, and exercise tolerance [20,21].

Evidence for early initiation of SGLT2i in acute HF settings has grown considerably, with results showing rapid-onset benefits and expanding their use to inpatient management of HF. The EMPULSE trial [22] showed that empagliflozin, started during acute HF hospitalization, improved clinical outcomes and reduced mortality within 90 days.

Similarly, Carvalho et al. showed reduced all-cause mortality and HF readmission in acutely decompensated patients receiving SGLT2i, without worsening renal dysfunction [23].

A core mechanism of this property of SGLT2i involves osmotic diuresis and natriuresis, reducing preload and afterload, which in turn reduces congestion in HF [23]. Enhanced diuresis also lowers loop-diuretic requirements, improving fluid management and reducing symptoms.

SGLT2i, due to its hypoglycaemic role, shifts myocardial substrate use toward ketone bodies, which are more energy-efficient in failing hearts [24]. Also, it reduces intracellular sodium by modulating the Na^+^/H^+^ exchanger (NHE1), thereby stabilizing intracellular calcium and reducing oxidative stress [25,26]. This “sodium-interactome” hypothesis combines improved contractility, reduced arrhythmogenesis, and lower myocardial workload.

Beyond hemodynamic benefits, many animal studies demonstrated slowed inflammatory pathways, including reduced levels of proinflammatory cytokines (IL-6, TNF-α) and inhibition of the NLRP3 inflammasome [24,25,26,27,28,29]. By modulating fibrosis and remodelling, SGLT2i interrupt inflammatory cycles that increase HF progression.

### 2.2. SGLT2i and Myocardial Infarction

Even more interesting is the use that this class of drugs would have in myocardial infarction (MI) patients in the post-episode period to prevent the reduction in ventricular function associated with the ischaemic event. The mechanism by which SGLT2i exert this effect is not completely clear but appears to be associated with metabolic changes related to the production and utilization of ketone bodies and in particular beta-hydroxybutyrate, which would appear to block NOD-like receptor protein 3-mediated inflammatory processes: this has been seen to correlate with a reduction in NT-proBNP levels and thus an improvement in left ventricular function [30].

Similarly, a Swedish study showed how early initiation of SGLt2i in patients who had had a myocardial infarction reduced rates of hospitalization for heart failure and all-cause mortality [31].

Finally, these data were also evaluated by two recent trials focusing on the population suffering from myocardial infarction. In particular, the DAPA-MI study showed a significant benefit that early treatment with dapagliflozin in MI patients without a diagnosis of diabetes mellitus or heart failure had a better recovery of ventricular function and a lower rate of hospitalization for heart failure [32]

Hernandez et al. [33] reported that empagliflozin significantly decreased first and total HF hospitalizations among patients with recent MI and left ventricular dysfunction or congestion. High-risk populations, including those with CKD or frailty, may benefit from these agents, with consistent reductions in HF hospitalizations and CV death [27,28].

### 2.3. SGLT2i and Aortic Stenosis

In the most recent literature and also in clinical practice, there has been a change in the conception of aortic stenosis (AS), which is no longer seen as a disease affecting only the valvular system, but a pathology that also involves the myocardium. For this reason, the evaluation of extra-valvular cardiac damage (EVCD) is crucial for the assessment and prognosis of patients with AS. So far, however, medical therapy has not proved effective in delaying disease progression, and for this reason, surgical therapy remains the most effective choice. Since pressure overload in severe AS leads to a metabolic imbalance in the heart with predominance of glucose metabolism over lipid metabolism and subsequent accumulation of lipids in the myocardiocytes, several studies have investigated if this behaviour can be reversed in consideration of the presence of increased expression of SGLT2 especially in patients with low flow, low gradient AS, hypothesizing that an early initiation of SGLT2i in these patients may improve the final outcome and remodelling [12].

The first study to indicate such an association is that of Paolisso et al. who demonstrated that the early initiation of SGLT2i in patients undergoing Transcatheter Aortic Valve Implantation (TAVI) leads to a better post-operative outcome with improved EVCD with an advantage that is not limited to the recovery of ventricular function but extends to an overall benefit of cardiac contractility up to one year after the initiation of therapy [34]. Furthermore, these beneficial results have also been proven in non-severe aortic stenosis in slowing progression to severe stenosis [35]. Finally, Raposeiras-Roubin et al. demonstrated how dapagliflozin significantly reduced the risk of death from all causes and worsening heart failure in elderly patients with AS undergoing TAVI at high risk of heart failure [36].

The effects of SGLT2i therapy on cardiovascular outcomes are summarized in Figure 1.

## 3. Effects of SGLT2i on Renal Outcomes

Several large clinical trials have demonstrated improvements in renal outcomes in high-risk patients with CKD and T2D. The nephroprotective effects of SGLT2is have been demonstrated in five major CVOTs, in which the renal endpoints were evaluated only as secondary outcomes, as well as in dedicated trials extended also to CKD patients without T2D.

The CVOT trials EMPA-REG Outcome, CANVAS, and DECLARE demonstrated, as secondary prespecified outcomes, a positive class-effect of the SGLT2i in significantly reducing albuminuria progression and preserving eGFR decline over time [37,38,39,40]. The CREDENCE study was the first trial specifically designed to assess renal outcomes in T2D patients with diabetic kidney disease. This study demonstrate that, when added to the standard nephroprotective treatment, in other words the maximum tolerated RAAS inhibition, canagliflozin exerted a hitting 30% risk reduction for the renal endpoint (dialysis for at least 30 days, transplantation, or a sustained eGFR of <15 mL/min/1.73 m^2^ for 30 days, doubling of the serum creatinine for at least 30 days, or death from renal or cardiovascular disease). Mainly, this result was independent of the baseline eGFR, and was also true for patients with more advanced CKD [39]. In addition, in the CREDENCE study, long-term risk reduction was also associated with significantly reduced albuminuria in the canagliflozin-treated group. The DAPA-CKD study extended these evidences to the more general population of CKD, with or without T2D with an eGFR of 25 to 75 mL/min/1.73 m^2^ and a urinary albumin-to-creatinine ratio of 200 to 5000 mg/g providing clinical evidence of the beneficial role of SGLT2is on the decline in the eGFR of at least 50%, end-stage kidney disease, or death from renal or cardiovascular causes, irrespective of the presence of T2D [40].

The EMPA-Kidney trial, including patients with chronic kidney disease who had an eGFR of at least 20 but less than 45 mL/min/1.73 m^2^, or who had an eGFR of at least 45 but less than 90 mL/min/1.73 m^2^ with a urinary albumin-to-creatinine ratio of at least 200 mg/g, showed that empagliflozin therapy led to a lower risk of progression of kidney disease or death from cardiovascular causes [41].

Mechanisms underneath these results are still not fully determined, although several pathways may contribute. SGLT2i decreases glucose reabsorption in the renal proximal tubules, thus promoting glucosuria. The degree of induced glucosuria is proportionally related to glycaemic control, as it is greater in patients with higher circulating glucose levels, as well as to the dose of drug administered [42]. The reduction of blood glucose levels is a significant pharmacodynamic effect of SGLT2. Moreover, this pattern of action is independent of insulin secretion, and this means that these drugs also work in patients with reduced pancreatic β-cell function. Fascinating, by blocking the SGLT2, these agents cause a decrease in sodium reabsorption. This leads to two important consequences: the natriuretic effect, which results in reduction of intravascular volume and BP [43], and the increase of sodium delivery to the macula densa that is followed by a reactivation of TGF, and a prompt reduction in single-nephron GFR. This effect is mediated by the vasoconstrictive adenosine and mitigates the renal hyperfiltration that is, in turn, responsible for deleterious long-term effects on the renal parenchyma [43].

SGLT2is have been shown to improve endothelial dysfunction and reduce oxidative stress and inflammation, all of which are largely related to the effect of glucose on renal vascular and tubular cells [44]. Furthermore, they have been shown to decrease the renal resistive index (RI), a marker of intrarenal vascular resistance that is also associated with a worse individual risk profile and prognosis.

Interestingly, SGLT2i seems to improve the regulation of extracellular matrix deposition, by decreasing the epithelial-to-mesenchymal transition, a mechanism that involves metalloproteinases and predisposes to renal fibrosis over time [45,46]. SGLT2is significantly lower albuminuria, thus reducing its toxic effects on renal tubules [47]. The reduction in albuminuria levels exerted by the SGLT2is is largely dependent on the reduction of intraglomerular pressure [48].

Furthermore, as is well known, mesangial cells and pericytes play a key role in regulating capillary blood flow and the passage of substances from the glomerular membrane. Several studies have reported the presence of SGLT2 in bovine retinal pericytes and mesangial cells [49]. Similarly, SGLT2 has also been shown to be present in renal proximal tubule cells, mesangial cells, and pericytes of non-diabetic mice [50]; and this could lead to a metabolic perturbation promoted by the mTORC1-signalling pathway that appears to be mitigated by SGLT2 inhibition [51].

In addition, the study by Li et al. showed how SGLT2 inhibition leads to increased podocyte protection in diabetes mellitus subjects via mitochondrial membrane-mediated energy balancing [52].

### SGLT2i and Contrast-Induced Acute Kidney Disease

One of the most frequent causes of acute renal failure is contrast media used in numerous diagnostic procedures and is therefore termed contrast-induced acute kidney disease (CI-AKI). This is often a transient injury that can have a negative prognosis from a cardiovascular and renal point of view in the short and long term [53]. Mechanisms by which iodinated contrast media can induce such damage appear to include bone marrow hypoxia, inflammation, and oxidative stress. There are several studies, including one on a Chinese population subjected to contrast media after angiography [54] and others on diabetic and non-diabetic populations who underwent contrast medium pre- and post-Percutaneous Coronary Intervention (PCI) with or without chronic renal failure, in which the use of SGLT2i reduced the occurrence, duration, and severity of CI-AKI episodes, positively impacting the prognosis of these patients [55,56,57].

## 4. Effects of SGLT2i on Electrolytic Balance and Uric Acid

It is important to be aware of the potential effects of SGLT2i on electrolytic balance and uric acid, especially in the common clinical setting of elderly T2D patients, with polypharmacy because of HF and/or CKD, often treated with RAAs inhibitors and diuretics, with a potential risk of electrolyte imbalance.

In healthy kidneys, the proximal tubule (PT) reabsorbs between 60 and 80% of the sodium loading that the glomeruli filter every day. The SGLT2 transporter plays a significant role in this process, reabsorbing one Na^+^ ion for every glucose molecule from the lumen of the early PT. In T2D, the higher expression and activity of SGLT2 and the complete recruitment of SGLT1 cause increased sodium reabsorption in PT, resulting in a reduction of sodium uptake at the level of the macula densa [58]. This pathway activates tubulo-glomerular feedback by a reduced synthesis of vasoconstrictive molecules acting on the afferent arteriola, leading to increased intraglomerular capillary hydrostatic pressure and finally hyperfiltration [59]. SGLT2 inhibition has the opposite effect, resulting in an increased delivery of sodium to the macula densa and a fall in intraglomerular pressure. However, available data in T2D patients suggest that enhanced natriuresis is transient, and the risk of hyponatremia is low.

T2D per se carries a higher risk of hyperkalaemia and is often associated with a variable degree of renal function impairment [60]. Furthermore, the frequent use of drugs affecting the RAAS may also contribute to the development of hyperkalaemia in diabetic patients [61,62]. In this regard, some studies have reported a potential and modest increase in serum potassium levels in patients treated with SGLT2i, namely in those treated with canagliflozin, but not with empagliflozin and dapagliflozin. In a pooled analysis of four placebo-controlled studies including canagliflozin-treated patients, the small increase in potassium levels was related to the drug dosage (100 versus 300 mg/day) and renal function (eGFR ≥60 mL/min/1.73 m^2^ versus eGFR ≥45 and <60 mL/min/1.73 m^2^) [63]. The incidence of serum potassium levels >5.4 mEq/L at any time during the follow-up was 4.5%, 6.8%, and 4.7% in patients treated with 100 mg and 300 mg of canagliflozin and placebo, respectively. In the EMPA-REG OUTCOME trial, the use of empagliflozin in patients with T2D and mild CKD (stage 2 and 3) was not associated with changes in serum potassium levels [64]. Similarly, dapagliflozin was not associated with serum potassium changes in patients with moderate renal impairment (eGFR 30 to 59 mL/min/1.73 m^2^) [64]. These encouraging results were also supported by a pooled analysis of 14 randomized clinical trials, which proved that dapagliflozin did not correlate with an increased risk of hyperkalaemia, even in patients with a baseline eGFR between 30 and 60 mL/min/1.73 m^2^ [65].

The effect of SGLT2i on kidney disease and on electrolyte balance are summarized in Table 1.

In addition, Tang et al. [66] in a meta-analysis of 18 randomized controlled trials, which included 15,309 patients with four SGLT2i (canagliflozin, empagliflozin, dapagliflozin, and ipragliflozin), showed that SGLT2i can increase serum magnesium levels by approximately 0.08 to 0.2 mEq/L in diabetic patients without normal renal function. Interestingly, canagliflozin increases serum magnesium in a linear and dose-dependent manner. These results are also confirmed by another analysis, which showed a dose-dependent increase in magnesium serum levels with increases in potassium and phosphate levels with canagliflozin administration. Of importance, these effects were larger in individuals with reduced renal function [67].

Additionally, a post hoc analysis based on pooled data from four randomized controlled trials (RCTs) of canagliflozin showed that more patients with baseline serum magnesium below <0.74 mmol/L presented a normal serum magnesium level at study completion, when treated with canagliflozin [68].

Interestingly, several evidence supported a beneficial effect of SGLT2i in reducing circulating levels of uric acid. SGLT2i induce a state of starvation mimicry at a molecular, cellular, and physiological level, which is characterized by glycosuria and ketogenesis [69] and by upregulation of nutrient deprivation signalling (i.e., SIRT1) and downregulation of nutrient surplus signalling (i.e., mTOR and HIF-1α) [70]. SGLT2 inhibitors slow glucose uptake by heart and kidneys [71] and reduce flux through the pentose phosphate pathway, thus redirecting glucose from anabolism to catabolism [72]. The two outcomes of this intracellular paradigm shift are: a decrease in the synthesis of purines; and decreased activity of NADPH oxidase. Therefore, SGLT2i can interfere with uric acid production by suppressing both purine synthesis and degradation.

In randomized controlled trials in patients with T2D or with HF, short- and long-term treatment with an SGLT2i lowers uric acid by ≈0.6–1.5 mg/dL, an effect that can be observed within the first week of therapy and persists for the duration of treatment [73]. The lowering effect of SGLT2i on serum uric acid is confirmed in patients with and without hyperuricemia, but it is greater in patients with baseline elevated serum uric acid concentrations [74].

In support of the above, in a Taiwanese national cohort, (47,405 patients with T2D who received an SGLT2i compared with 47,405 individuals matched for propensity score who received dipeptidyl peptidase 4 inhibitors), the use of SGLT2i in T2D was associated with a lower risk of gout incidence than the use of DPP4 inhibitors [75]. In this study, the incidence of gout was reduced by 11% with SGLT2i [75].

## 5. Effects of SGLT2i on Body Weight

SGLT2i therapy has been associated with a variable and time-limited body weight-lowering effect.

The caloric loss/energy deficit determined by the glycosuria brought about by the treatment with SGLT2i has been the result of approximately 300 Kcal per day. This should lead to an expected weight loss at 24 weeks of treatment of about 7 kg. Nevertheless, clinical trial data suggest SGLT2i produces a mean weight loss of approximately 2 to 3 kg in patients with T2D, irrespective of the background therapy [76].

In a phase II study, the use of Canagliflozin produced a weight loss of approximately 2–3 kg compared to baseline, and of approximately 1–1.5 kg compared to placebo in overweight or obese patients with or without diabetes, over a 12-month period [77]. In real-world data, one observational study demonstrated a mean weight loss of 2.6 kg from 14 to 90 days after starting dapagliflozin and 4.6 Kg beyond 180 days [78].

There is great interest as to why SGLT2i therapy is not associated with more pronounced weight loss, in consideration of the caloric loss/energy deficit. It has been suggested that the discrepancy between the observed and predicted weight loss with SGLT2i may result from compensatory increases in energy intake and changes in energy expenditure that act to mitigate the energy imbalance. Furthermore, it has been demonstrated that the increase in urinary glucose excretion following SGLT2 inhibition is associated with a paradoxical increase in endogenous (hepatic) glucose production, possibly caused by a compensatory release of glucagon by α-cells in the pancreatic islets, leading to a partial reversal of the metabolic effect of SGLT2i [2]. Moreover, human studies have shown that a 4-week administration of empagliflozin causes a shift in fuel utilization from carbohydrate to fatty substrates, leading to a loss of fat mass and weight, but no changes in resting or postprandial energy expenditure, measured by indirect calorimetry, were observed, implying an increase in energy intake to explain the discrepancies in observed weight loss [79].

## 6. Effects of SGLT2i on Blood Pressure

Arterial hypertension and T2D are frequently co-existing diseases: approximately 50 per cent of diabetic patients have hypertension at the time of diagnosis, and the prevalence increases with the duration of the disease. Although SGLT2i have not been approved as antihypertensive drugs, there is growing evidence on their positive effect on blood pressure (BP).

The EMPA-REG BP study aimed to investigate the efficacy of empagliflozin on 24-h blood pressure monitoring and clinical BP [80] in BP reduction was found in the active treatment arm, irrespective of the dose of empagliflozin [81].

Other studies have focused on patients with HF. Although they showed promise in improving cardiovascular and renal prognosis, the benefits of BP mediated by SGLT2i were less pronounced. The DAPA-HF trial showed a reduction in systolic blood pressure of 1.27 mmHg with dapagliflozin 10 mg. Similarly, EMPA-Reduced demonstrated a reduction in systolic blood pressure of only 1–2 mmHg. The effect was similar in the EMPEROR trial, the results of which suggested that empagliflozin was associated with a reduction in systolic BP of 1.8 mmHg. A recent meta-analysis of 16 randomized controlled trials involving patients with HF, with and without T2D, showed that SGLT2 inhibition was associated with a modest but significant reduction in systolic BP (1.68 mmHg), but not in diastolic BP (1.06 mmHg). In patients with HF, the antihypertensive effect of SGLT2i was minimal and transient [82].

In hyperglycaemic conditions, SGLT2 activity is increased, enhancing its ability to reabsorb glucose in the proximal tubule in response to elevated plasma glucose levels. This hyperactivity of SGLT2 also results in increased sodium reabsorption, a mechanism that may contribute to the pathogenesis of hypertension. With SGLT2 inhibition, sodium reabsorption is reduced, producing a mild diuretic effect, which contributes to short-term pressure reduction [83].

Weber conducted a randomized, double-blind, placebo-controlled study examining the effect of dapagliflozin on BP in patients with T2D and normal renal function, already treated with antihypertensive combination therapy. A reduction in systolic blood pressure of 4.3 mmHg was demonstrated in patients receiving dapagliflozin 10 mg [84]. A post hoc analysis showed that dapagliflozin in addition to beta blockers or calcium antagonists led to a marked reduction in systolic BP, of 5.7 mmHg and 5.1 mmHg, respectively [85]. In contrast, patients already treated with diuretics did not achieve a further reduction in BP with the add-on of SGLT2i [86]. This evidence suggests that SGLT2i reduces blood pressure by volume reduction [87].

Several experimental studies have also demonstrated that the BP lowering effects of SGLT2 inhibition may be mediated by mechanisms unrelated to glycosuria: e.g., by reducing arterial stiffness [88]; to an improvement in endothelial dysfunction [89] or the influence they would have on the activity of endothelial nitric oxide synthase, in particular endothelial nitric oxide synthase, by remodulating NO synthesis and normalizing vascular tone. In addition, SGLT2i counteract oxidative stress by reducing the production of free radicals [90].

Notably, in most patients with T2D, hyperactivity of the sympathetic nervous system has been observed, and SGLT2i inhibition could modulate the autonomic nervous system [73]. Despite a reduction in BP and plasma volume, heart rate does not increase with the use of SGLT2i, suggesting a reduction in cardiac sympathetic tone and an increase in parasympathetic tone [91].

In various studies, a weight loss of 1 to 3 kg has been demonstrated in patients treated with SGLT2i, and this seems to have a positive impact on cardiovascular risk factors, including better BP control [88].

## 7. Effects of SGLT2i on Lipid Profile

T2D is associated with a typical form of dyslipidaemia characterized by high triglyceride concentrations, a high prevalence of small and dense low-density lipoprotein (LDL), and low concentrations of high-density lipoproteins (HDL). This highly atherogenic lipid dysregulation leads to an increased risk of cardiovascular disease [92]. SGLT2i, first developed as hypoglycaemic drugs, have also been shown to reduce mortality from cardiovascular events. Over the past decade, there has been a growing focus on evaluating the effects of SGLT2 inhibition on the concentration of various plasma lipoproteins to recognize their possible influence in the demonstrated reduction of cardiovascular risk. A meta-analysis of 60 randomized placebo-controlled studies involving a total of 147,130 individuals analyzed the effects of different available drug doses, considering the lipid profile closest to 52 weeks of follow-up. Analysis of the data showed that treatment with SGLT2i increases total cholesterol, LDL-cholesterol, and HDL-cholesterol levels and reduces triglyceride concentrations [93]. These outcomes are in contrast to the evidence that SGLT-2i, in addition to reducing cardiovascular risk, affect body composition in individuals with T2D by reducing body mass index, visceral fat, subcutaneous fat, fat mass, and body fat percentage [94]. The consequences of SGLT2 inhibition on lipid profile have been analyzed in a meta-analysis that included 42 randomized controlled clinical trials involving the administration of the highest approved therapeutic dose of canaglifozin, dapaglifozin, and empaglifozin for at least 12 weeks, with the greatest effects observed for canaglifozin on the increase in LDL and HDL cholesterol and the reduction of triglycerides [95]. An experimental study was conducted to evaluate the molecular mechanisms beyond lipid modifications with SGLT2i. A diabetic mouse model expressing human CETP (cholesteryl ester transfer protein) and human ApoB100 (apolipoprotein B100) was used to try inhibiting SGLT2 via a specific antisense oligonucleotide or canagliflozin and reproduced many of the lipid changes seen in humans during treatment with SGLT2i, the increase in LDL-cholesterol, and the reduction in triglycerides. The results showed an increase in lipoprotein kinase (LpL) activity, which would explain the increased serum concentration of LDL lipoproteins with the increased conversion of very high-density lipoproteins (VLDL) to LDL, possibly also associated with the increase in HDL cholesterol. Secondarily to the increase in lipolytic activity induced by SGLT2 inhibition, a reduction in postprandial lipaemia was also observed, which correlated with a reduced risk of cardiovascular disease [93], a decrease in circulating triglyceride levels, and a delayed clearance of LDL lipoproteins. The study’s further revelation concerns a post-transcriptional mechanism induced by SGLT2 inhibition, resulting in reduced gene expression of ANGPTL4 (Angiopoietin-like protein 4), which is capable of inactivating LpL, thus enabling its increased activity [96].

Genetic studies have shown an association between reduced ANGPTL4 gene expression and reduced CVD risk [97,98,99]. Further data supporting the role of lipolysis and triglycerides in cardiovascular risk variation come from large-scale genetic studies that have identified loci associated with triglycerides, LDL and HDL cholesterol, lipolytic activity, and CVD [100]. This evidence suggests that possible mechanisms underlying the lipid alterations induced by SGLT2i in humans derive from the observed increase in lipolytic activity and suggest that the cardiovascular protection demonstrated in treatment with these drugs may be the outcome of factors arising from ANGPTL4 inhibition.

## 8. Effects of SGLT2i on Bone Metabolism

In recent years, the relationship between T2D and bone metabolism has become an important area of research and clinical development, starting from the observation that T2D patients have an increased fracture risk, in spite of a preserved bone mass [101]. Moreover, T2D itself and its long-term complications (retinopathy, neuropathy) as well as several hypoglycaemic drugs may contribute to either the risk of falls or fractures, as well as to the impairment of bone health [102].

T2D is associated with impaired or even suppressed bone turnover [103,104]. However, from available evidence, the role of SGLT2i on bone metabolism has not been fully elucidated yet.

In murine models, treatment with canagliflozin was associated with a significant increase in bone resorption markers, in particular carboxy-terminal telopeptide of collagen type 1 (CTX) and a marked deficits in cortical and trabecular bone architecture, with an increase in calcinuria and circulating serum levels of Fibroblast Growth Factor 23 (FGF-23) with evident deterioration of trabecular bone mass [105,106]. Furthermore, the Slc5a2 nonsense mutation results in total loss of SGLT2 function in mice, this, in addition to leading to increased glycosuria and normalized renal function, resulted in reduced urinary calcium and phosphorus excretion, with normal serum calcium and phosphate levels, normal serum levels of parathormone (PTH), vitamin D and FGF-23. In contrast, reduced femoral length with reduced bone mineral density was noted at 25 weeks in these mice, suggesting skeletal impairment independent of circulating levels of the major bone turnover markers [107].

Various studies showed that SGLT2i may increase serum phosphate levels by reducing sodium transport. In human studies, phloridzin (an old non-selective SGLT1/SGLT2 inhibitor) reduced phosphate clearance by an average of 80% during the first hour after its administration [108]. Similarly to phlorizin, SGLT2i increased serum phosphate levels, augmenting PTH secretion by the parotids, and leading to increased FGF-23 secretion by osteocytes. PTH and FGF23 exert opposite effects on the 1α-hydroxylation of 25-hydroxyvitamin D. While the former increases its levels, the latter decreases its hydroxylation [109]. Consequently, if SGLT2i increases both PTH and FGF-23 levels, the net impact on 1,25-dihydroxyvitamin D could not be predicted a priori.

In line with these mechanisms, Bilezikian et al. [110] observed that increased bone turnover, as demonstrated by increased CTX and osteocalcin levels. In contrast to these data, it has been reported that dapagliflozin does not influence bone mineral density or biomarkers of bone turnover in patients with normal to mildly impaired renal function [111]. Although these contrasting data may suggest that SGLT2i effects on bone turnover are molecule-specific, other data suggest that any differences between the compounds may be the consequence of dose selection rather than an intrinsic difference of the compounds [112].

In a real-world study, canagliflozin resulted in a reduction of bone mineral density (BMD) of the hip in a cohort of patients aged 55 to 80 years, and the change in body weight seemed to explain about 40% of the observed difference [110]. In another study, the reduction of estradiol levels seemed to partly contribute to the loss of BMD, also an effect that seems to be related to canagliflozin treatment [113]. In contrast, Rosenstock et al. reported that ertugliflozin had no adverse effects on BMD in the lumbar spine, femoral neck, hip, and distal forearm regions after 26 weeks of treatment in either the overall population or in the cohort of postmenopausal women [114]. Similarly, dapagliflozin showed no significant differences in BMD in the lumbar spine, femoral neck, and hip after 50 and 102 weeks of treatment [111].

On the other hand, in a meta-analysis of 78 randomized controlled trials, among all the SGLT2i, only treatment with canagliflozin was associated with an increased incidence of fractures [115]. The CANVAS study (CANagliflozin cardioVascular Assessment Study Program) revealed a higher risk of low-trauma fractures and of all fractures in the canagliflozin group compared to the placebo group [116], but the CANVAS-R study did not confirm this observation [116]. To date, there is no clear explanation for the observed differences between these two studies, which included comparable patient groups and evaluated the same intervention, and therefore, the reason for the increased risk of fractures observed in the CANVAS study remains unknown [117].

In contrast, in two recent meta-analyses examining the impact of SGLT2i on BMD, bone metabolism markers, and fractures, no statistically significant worsening of any bone parameter was observed [118].

Only further long-term studies in different clinical settings may elucidate the effects on bone metabolism of this class of drugs.

## 9. Effects of SGLT2i on Liver

Metabolically related liver disease and T2D are highly connected and frequently observed in clinical practice [119]. Accordingly, metabolic dysfunction-associated liver disease (MASLD) has recently been proposed to replace the term non-alcoholic hepatic steatosis (NAFLD) [120]. This change considers the pathogenesis of the disease, which is closely linked to metabolic disorders such as T2D, obesity, and metabolic syndrome [121].

SGLT2i appear to have beneficial effects on hepatic steatosis, and they have been extensively studied as a potential therapeutic option for MASLD.

According to the literature, all commercially available SGLT2i lower serum levels of alanine aminotransferase (ALT), aspartate aminotransferase (AST), and gamma-glutamyl transferase (GGT), irrespective of the specific molecule used [122].

Additionally, these benefits extend to non-invasive assessments of hepatic steatosis and measurements obtained via transient elastography, such as the controlled attenuation parameter (CAP) and liver stiffness (LS) [123]. In the study by Lai et al., treatment with empagliflozin for 24 weeks led to a notable decrease in body mass index (BMI), waist circumference, liver fat volume fraction, and biopsy-confirmed steatosis, ballooning, and fibrosis compared to placebo [124].

Dapagliflozin has also demonstrated efficacy, reducing hepatic steatosis as measured by hepatic proton density fat fraction (PDFF) on MRI after 6 months of therapy and via histological evaluations [125]. In terms of liver function tests, decreases in transaminases and γGT at 6 and 12 months have been reported [126], although serum bilirubin levels may increase [127]. Among the SGLT2i agents, dapagliflozin specifically showed a more pronounced reduction in the Fatty Liver Index (FLI) after 12 weeks [128] and in the Visceral Adipose Index [129] compared to pioglitazone. However, when dapagliflozin was directly compared to exenatide—whether administered alone or in combination—FLI declined in groups at 28 weeks, whereas FIB-4 decreased only in the group receiving both therapies [130].

Comparisons of SGLT2i with dipeptidyl peptidase-4 inhibitors (DPP4i) have produced mixed findings. One study showed no differences in major liver outcomes, including decompensated cirrhosis, variceal bleeding, ascites, and hepatic encephalopathy [131]. Another study showed that SGLT2i is superior to DPP4i in reducing indices of hepatic inflammation and fibrosis, and in suppressing the incidence of oesophageal varices and extrahepatic cancer [132].

## 10. SGLT2i and Cancer

SGLT2 inhibitors exert a broad spectrum of biological effects and have emerged as promising agents in oncology for both diabetic and non-diabetic populations. Due to their distinct mechanism of action, they are considered a compelling therapeutic option for mitigating cardiovascular events and enhancing overall survival in cancer patients with pre-existing cardiovascular comorbidities or those undergoing cardiotoxic treatments [133]. Evidence from both preclinical and clinical studies supports the cardioprotective role of SGLT2 inhibitors in managing cancer therapy-related cardiovascular toxicity (CTR-CVT), particularly that associated with anthracyclines, immune checkpoint inhibitors, and HER2-targeted agents. These benefits include the prevention of left ventricular ejection fraction decline and stabilization of cardiac electrophysiological parameters [134]. The cardioprotective effects in oncologic settings appear to involve multiple pathophysiological mechanisms, including modulation of macrophage phenotype, regulation of cardiomyocyte Ca^2+^/Na^+^ homeostasis, inhibition of inflammasome activation, normalization of myocardial SGK1 and ENaC expression, increased production of ketone bodies, and suppression of ferroptosis [133]. Ongoing clinical trials, such as Empaglifozin in prevention of chemotherapy-related cardiotoxicity (EMPACT/NCT05271162), will give additional answers whether prophylactic SGLT2i may prevent a reduction in LVEF after high-dose anthracyclines [135].

Nevertheless, SGLT2i have demonstrated potential antitumor effects in both in vitro and in vivo studies across various solid and hematologic malignancies [133,136,137].

Their mechanism of action involves impairing glucose uptake in tumor cells by targeting SGLT1, SGLT2, or GLUT1 transporters, disrupting cellular energy production, and triggering AMPK/mTOR signaling, which suppresses cell growth [138].

These drugs also influence tumor metabolism by reversing the Warburg effect, a hallmark of tumor metabolism characterized by enhanced glycolysis, and inhibiting hexokinase II activity, a key enzyme in this process. Beyond metabolic interference, SGLT2 inhibitors demonstrate immunomodulatory effects by promoting PD-L1 degradation and enhancing T-cell infiltration, as well as reducing inflammation and oxidative stress. Other effects include mitochondrial membrane instability, suppression of β-catenin and PI3K-Akt pathways, an increase in cell cycle arrest and apoptosis [139].

Some evidence, not only from preclinical models, but also from retrospective studies and meta-analysis, suggests these agents—particularly canagliflozin, dapagliflozin, and empagliflozin—may reduce tumor growth and proliferation in cancers such as gastrointestinal [140], colorectal [141], liver [142], lung [143], breast [144], prostate [145], and pancreatic malignancies [146]. Notably, their efficacy may vary depending on tumor metabolic phenotype and stage; for instance, SGLT2 seems to be more represented in precancerous lesions and early, well-differentiated lung adenocarcinoma in mice, suggesting that SGLT2 is a diagnostic and therapeutic target with a greater benefit for early-stage lung adenocarcinoma than in advanced disease [136,147]. Ongoing trials are being led to evaluate their safety and feasibility as part of combination therapies with established antitumor agents [137]. Although promising, further research is needed to determine the specific cancer types and molecular contexts in which SGLT2 inhibitors offer the greatest therapeutic advantage.

## 11. Future Perspectives: IBD, Cognitive Impairment, and COPD

SGLT2i have been investigated in other areas for potential application, including chronic inflammatory bowel disease (IBD) and cognitive impairment.

Thanks to the anti-inflammatory properties of SGLT2i, it is not surprising that a growing amount of literature is aiming to evaluate a possible role of SGLT2i in different inflammatory diseases, such as chronic IBD. Specifically, Makaro et al. [114] tested empagliflozin, dapagliflozin, and canagliflozin in in vitro and in vivo models of intestinal inflammation. In vitro experiments revealed that both empagliflozin and dapagliflozin suppress the production of nitric oxide and alleviate acute DSS-induced colitis in mice. Therefore, treatment with empagliflozin reduced macro- and microscopic colonic damage, as well as partially prevented from decrease in tight junction gene expression and attenuated biochemical inflammatory parameters, including reduced activity of myeloperoxidase [148]. These observations suggest that gliflozins alleviate inflammation through their potent effects on innate immune cells. In another study, Canagliflozin attenuates inflammation in AA-induced colitis, evidenced by significant and dose-dependently downregulation of p38 MAPK, NF-κB-p65, IKK, IRF3, and NADPH-oxidase as well as colonic levels of IL-6 and IL-1β and MPO enzymatic activity [149]. Finally, dapagliflozin (0.1, 1 and 5 mg/kg; p.o.) dose-dependently mitigated colitis severity in 2,4,6 trinitrobenzene sulfonic acid (TNBS)-induced rat colitis model as manifested by suppression of the disease activity scores, macroscopic damage scores, colon weight/length ratio, histopathologic perturbations, and inflammatory markers [150].

Another vast chapter in T2D management is addressing cognitive impairment, also considering the progressive ageing of T2D subjects over time. Thus, according to updated data from a large population in Italy, >30 T2D patients attending outpatient clinics are over 75 years (Russo GT AMD Annals 2023 DRCP 2024). Subjects suffering from T2D are at higher risk of vascular disease, cognitive decline, and dementia, opening up to a growing scientific research focusing on the use of SGLT2i in the prevention of dementia in patients with T2D. Contradictory results emerge from two of the meta-analyses available in the literature: in the one conducted by Youn et al., the use of SGLT-2i significantly lowers the risk of dementia compared to SGLT2i non-users, as shown on cognitive function score [151]. On the other side, the meta-analysis by Jaiswal et al. did not show any significant association between SGLT2i use and risk of dementia and Parkinson’s disease [152].

In a Korean nationwide population-based study, in which 358,862 subjects were involved, SGLT2i use was associated with reduced risks of Alzheimer’s Disease, vascular dementia, and Parkinson’s Disease with a 6-month drug use lag period. In addition, use of SGLT2i was associated with a 21% lower risk of all-cause dementia and a 22% lower risk of all-cause dementia and PD than use of other oral antidiabetic drugs. The association between the use of SGLT2i and the lowered risk of these neurodegenerative disorders was not affected by sex, diabetic complications, comorbidities, and medications [153].

In two comparative studies, SGLT2i were associated with a reduced risk of all-cause dementia compared to sulfonylureas [154] and, in a population of 106,903 Ontario residents aged ≥66 years, treatment with SGLT2i was associated with a lower risk of dementia over a mean follow-up of 2.80 years, when compared with DPP-4 inhibitors. In this study, however, when stratified by different SGLT2i, dapagliflozin exhibited the lowest risk, followed by empagliflozin, whereas canagliflozin showed no association [155].

Chronic obstructive pulmonary disease (COPD) is an adverse independent prognostic factor in patients with HF and in diabetic individuals. SGLT-2i, despite lacking pulmonary receptor expression, may provide respiratory benefits in COPD patients by reducing CO_2_ retention via glucosuria and lowering pneumonia risk. Their glucosuric effect decreases serum glucose, subsequently reducing endogenous carbon dioxide production, which is beneficial for individuals with impaired CO_2_ clearance [156]. Additional mechanisms include enhanced oxygen delivery through increased erythropoietin [157,158], reduced pulmonary congestion via diuresis, especially in heart failure patients, resulting in improved alveolar perfusion [159], decreased airway hyperresponsiveness and fibrosis [160], and suppression of systemic inflammation through cytokine reduction and NLRP3 inflammasome inhibition [161,162]. Moreover, SGLT-2 inhibitors mitigate pulmonary hypertension and obstructive sleep apnoea, suggesting broader respiratory advantages [163,164,165].

To the best of our knowledge, no data are available on the efficacy of SGLT-2 inhibitors in COPD patients without type 2 diabetes or heart failure. A meta-analysis of 9 CVOTs by Yin et al. linked SGLT-2i to a reduced risk of 11 respiratory diseases, including COPD [166]. A population-based cohort study found a 38% decreased risk of severe COPD exacerbations, though not moderate ones, compared with sulfonylureas in patients with COPD and T2D [167], whilst benefits on mild-to-moderate COPD patients were observed in a pre-specified analysis of DELIVER [168]. A cohort study from the Australian Diabetes Registry associated SGLT-2 inhibitors with reduced hospitalizations for obstructive airway diseases [169], aligning with findings from Au et al., which demonstrated a lower exacerbation rate compared to DPP-4 inhibitors [170]. A recent Taiwanese cohort study further reported reduced COPD exacerbations, ventilatory support need, and mortality with SGLT-2 inhibitor use, with higher cumulative doses and treatment duration correlating with lower hospitalization risk [171]. While this study did not find a significant reduction in cardiovascular events or heart failure hospitalizations, post hoc analyses of CVOTs demonstrated that dapagliflozin and empagliflozin consistently lowered cardiovascular event risk, heart failure hospitalizations, and all-cause mortality in patients with and without COPD [172,173].

## 12. Euglycemic Ketoacidosis and Post-Surgery

SGLT2i are known to increase the risk of euglycemic diabetic ketoacidosis. This complication is characterized by hyperglycemia, increased blood ketone concentrations, and metabolic acidosis, which are related to absolute or relative insulin deficiency and an increase in counter-regulatory hormones [174]. Surgery and prolonged fasting may precipitate the development of SGLT2 inhibitor-associated ketoacidosis [175], and a systematic review of perioperative SGLT2 inhibitor-associated ketoacidosis revealed that 89% of patients presented with euglycemic ketoacidosis (blood glucose concentration of <250 mg/dL) [176]. The most common symptoms at presentation are nausea, vomiting, tachypnea, and abdominal pain, although postoperative patients without ketoacidosis also commonly experience these symptoms. Therefore, it can be challenging to diagnose SGLT2 inhibitor-associated ketoacidosis in surgical patients. Underlying this increased risk seems to be the action of the counter-regulatory hormones. These hormones help create a milieu of insulin resistance, which releases the brakes off lipolysis. Degradation of adipose tissue triglycerides leads to a massive efflux of free fatty acids (FFAs) to the liver, which is the main site of conversion of FFAs to ketones. On the other hand, in classic DKA, patients also develop hyperglycemia due to accelerated glycogenolysis and an increase in gluconeogenesis [174]. Despite insulin resistance and relative insulin deficiency, patients on SGLT-2 inhibitors are generally normoglycemic or moderately hyperglycemic. This phenomenon is likely related to constant glycuresis achieved by co-transporter inhibition in the kidneys or decreased glucose production by the liver during the fasting state.

## 13. Conclusions

In conclusion, according to the data presented here, it can be summarized how the increasing use of SGLT2i in T2D patients and non-diabetic patients is justified by their important metabolic, cardio- and nephro-protective role. Their ability to act on multiple levels, providing the patient with increased protection from most modifiable risk factors of cardiovascular and renal disease (Table 2), places this class of drugs among the most prescribed at diabetes clinics. Although present, side effects are easily predictable and treatable, making these molecules safe and reliable. The pleiotropic multi-level effects also open further avenues towards a multidisciplinary use of these molecules (Figure 2).

## Figures and Tables

**Figure 1 cells-14-00668-f001:**
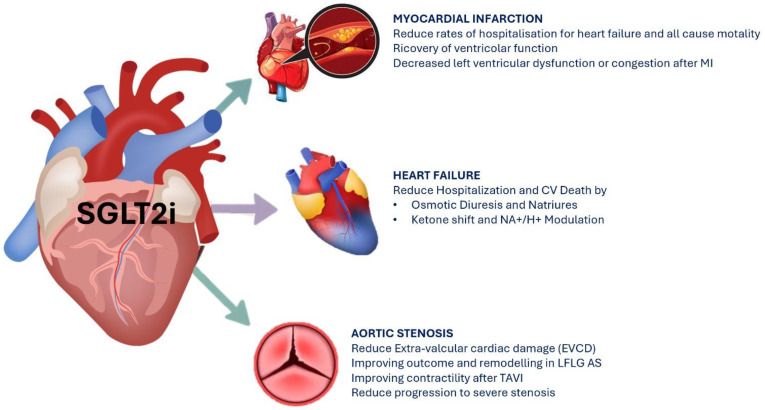
SGLT2i effects on cardiovascular outcomes.

**Figure 2 cells-14-00668-f002:**
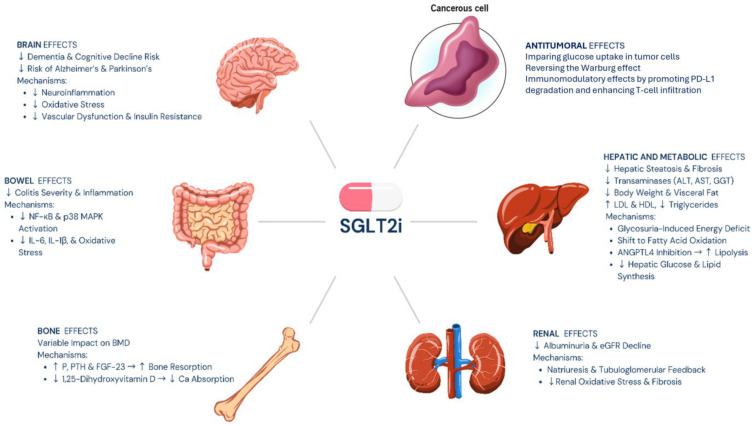
Potential beneficial mechanisms of SGLT2i in different clinical settings.

**Table 1 cells-14-00668-t001:** Effect of SGLT2i on kidney disease and electrolytic balance.

	Mechanism	Effect	Ref. No.
eGFR	Decrease in sodium reabsorption which results in reduction of intravascular volume [43] and the increase of sodium delivery to the macula densa that is followed by a re-activation of TGF. This effect is mediated by the vasoconstrictive adenosine and mitigates the renal hyperfiltration that is in turn responsible for deleterious long-term effects on the renal parenchyma [43]. Improve endothelial dysfunction and reduced oxidative stress and inflammation and decrease the renal resistive index (RI) [44].	preserving eGFR decline over time	[37,38,39,40]
		↓ end-stage kidney disease, or death from renal or cardiovascular causes	[39,40,41]
		↓ 30% risk for dialysis for at least 30 days, transplantation, or a sustained eGFR of <15 mL/min/1.73 m^2^ for 30 days	[37,38,39,40]
Albuminuria	Improve the regulation of extracellular matrix deposition, by decreasing the epithelial-to-mesenchymal transition by metalloproteinase [45,46]. Increased podocyte protection via mitochondrial membrane-mediated energy balancing [52].	↓ albuminuria	[37,38,39,40,41]
Na^+^ balance	Increased delivery of sodium to the macula densa and a fall in intraglomerular pressure [58].	↑ transient natriuresis with low risk of hyponatremia	[59]
K^+^ balance		small increase in potassium levels only with canagliflozin	[63]
Mg^+^ balance		↑ serum magnesium levels by approximately 0.08 to 0.2 mEq/L	[66]

**Table 2 cells-14-00668-t002:** Control of major cardiorenal risk factors.

Risk Factor	Direction (↑/↓)	Range/Δ (Units)	Ref. No.
BMI/body weight	↓	−1.9 → −4.6 kg (≈−0.66 → −1.6 kg m^−2^)	[76,77,78]
Blood pressure	↓	−1.3 → −5.7 mmHg (systolic)	[80,84]
LDL-cholesterol	↑	+0.07 → +0.12 mmol L^−^^1^ (+2.7 → 4.6 mg dL^−^^1^)	[93]
HDL-cholesterol	↑	+0.05 → +0.08 mmol L^−^^1^ (+1.9 → 3.1 mg dL^−^^1^)	[93]
Triglycerides	↓	−0.13 → −0.07 mmol L^−^^1^ (−11.5 → −6.2 mg dL^−^^1^)	[93]
Uric acid	↓	−0.53 → −1.54 mg dL^−^^1^ (−31 → −91 µmol L^−^^1^)	[73,74]

## Data Availability

All data analysed in this study are included in this published article.

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
