# Peer review of "Expanding the Use of SGLT2 Inhibitors in T2D Patients Across Clinical Settings"

_cells, 2025, doi:10.3390/cells14090668_

Round 1
Reviewer 1 Report
Comments and Suggestions for Authors
Two important fields should be added to the review :
1- The Anticancer activity of SGLT-2 inhibitors due to the well-known fact that cancer cells typically switch their metabolism from fatty acid oxidation to glucose utilisation This is because the cancer cells often contain dysfunctional mitochondria which do not provide enough energy from the metabolism of the fatty acids. Moreover incidence of cancer is incresed in type 2 patients .
2- Ketogenesis, why and what implication before surgery...
Author Response
We would first like to thank the Referee for the appreciation to our work, and for the valuable suggestions, which we have responded point by point, as specified below.
Two important fields should be added to the review :
1- The Anticancer activity of SGLT-2 inhibitors due to the well-known fact that cancer cells typically switch
their metabolism from fatty acid oxidation to glucose utilisation This is because the cancer cells often contain
dysfunctional mitochondria which do not provide enough energy from the metabolism of the fatty acids.
Moreover incidence of cancer is incresed in type 2 patients.
Re: Anticancer activity of SGLT2i is an interesting and expanding field of research, both for the protective effects on cardiomiotoxicity of chemioterapics and for direct protective effects on cancer growth. Accordingly, a whole new paragraph on this topic has been added to the revised version of our review. Please see paragraph n. 10- SGLT2i and cancer.
2- Ketogenesis, why and what implication before surgery...
Re: The risk of ketoacidosis associated with the use of SGLT2i is a well-known aspect, that our group recently reviewed in Morace C, Lorello G, Bellone F, Quartarone C, Ruggeri D, Giandalia A, Mandraffino G, Minutoli L, Squadrito G, Russo GT, Marini HR. Ketoacidosis and SGLT2 Inhibitors: A Narrative Review. Metabolites. 2024 May 6;14(5):264. doi: 10.3390/metabo14050264. PMID: 38786741; PMCID: PMC11122992. However, the risk of ketoacidosis in patients undergoing surgery is a less known and clinically relevant topic, that could be omitted in the perioperative evaluations of patients. As suggested, we added a paragraph on this specific topic in the revised
version of our manuscript. Please see paragraph 12. Euglycemic Ketoacidosis and post-surgery.
Reviewer 2 Report
Comments and Suggestions for Authors
Cuttone A. et al. explored the evidence accumulating on the potential effects of SGLT2i in different clinical conditions, beyond glycemic control.
The review is up-to-date and of interest to the readers; nevertheless, I have some concerns that need to be addressed before the study can be re-submitted.
- Are the authors discussing the clinical advantage of SGLT2i in different clinical settings commonly encountered in patients with T2D, or are these benefits also extended in patients without diabetes mellitus? Please clarify.
- I would suggest shortening the first part of the introduction regarding the pathophysiology of diabetes mellitus, as it seems beyond the scope of the review. Consider beginning with the recommendations for SGLT2i use in patients with diabetes.
- Following the Introduction, I suggest including a paragraph discussing the ongoing debate regarding the expression of SGLT2i at the cardiac level (in cardiomyocytes), highlighting both supporting and opposing evidence (opposing: PMID: 32613148, 20045149; supporting: PMID: 30291295, 36096423, 38247224).
- Given the growing body of evidence in this field, it may be useful to split the section on cardiovascular outcomes into two distinct parts: “Effects of SGLT2i on Acute and Chronic Heart Failure” and “Effects of SGLT2i on Acute Myocardial Infarction.” Both sections should be expanded and updated with recent references (PMID: 39262451, 36036746).
- In the section “Effects of SGLT2i on Renal Outcomes,” consider adding a few sentences about the beneficial effects of SGLT2i in preventing contrast-induced acute kidney injury (CI-AKI), both in patients with chronic coronary syndrome and in those presenting with acute myocardial infarction.
- Please include a new section dedicated to the most recent and emerging evidence on the benefits of SGLT2i in patients with aortic stenosis undergoing transcatheter aortic valve implantation (TAVI), with and without diabetes (PMID: 39574095, 40162639, 38247224, 39985508).
- Kindly revise Figure 1 to reflect these suggested changes.
Author Response
Re: We would first like to thank the Referee for the appreciation, for the timely review of our work, and for the valuable suggestions, that have contributed to greatly improving it. We have welcomed each suggestion, and we responded point by point as specified below.
Cuttone A. et al. explored the evidence accumulating on the potential effects of SGLT2i in different clinical conditions, beyond glycemic control.
The review is up-to-date and of interest to the readers; nevertheless, I have some concerns that need to be addressed before the study can be re-submitted.
- Are the authors discussing the clinical advantage of SGLT2i in different clinical settings commonly encountered in patients with T2D, or are these benefits also extended in patients without diabetes mellitus? Please clarify.
Re: We agree with the Referee and in order to better clarify this issue, we have modified the title of the review as follows: “Expanding the Use of SGLT2 Inhibitors in T2D patients across Clinical Settings.”
- I would suggest shortening the first part of the introduction regarding the pathophysiology of diabetes mellitus, as it seems beyond the scope of the review. Consider beginning with the recommendations for SGLT2i use in patients with diabetes.
Re: We would like to thank the Reviewer for this suggestion. Accordingly, we have modified the introduction section, deleting the paragraphs on the pathophysiology of diabetes and other treatments and by adding a paragraph on current ADA recommendations for SGLT2i in T2D treatment.
Please see Introduction section of the revised manuscript, as follows:
“Uncontrolled T2D leads to the development of long-term micro- and macrovascular complications. T2D patients have a two- to three-fold increased risk of cardiovascular disease (CVD), which is further amplified in the presence of chronic renal impairment. In addition to atherosclerotic CVD, patients with T2D have an increased risk of developing diabetic kidney disease (DKD) and heart failure (HF) (3).
The initial approach recommended by current guidelines includes lifestyle changes and monotherapy, preferably metformin. Several other classes of drugs are currently recommended to control blood glucose, including glucagon-like peptide 1 (GLP1) receptor agonists, SGLT2 inhibitors (SGLT2i), thiazolidinediones, dipeptidyl peptidase-4 (DPP-4) inhibitors, and insulin. These drugs have achieved significant clinical success.
SGLT2i are FDA-approved for managing adult patients with type 2 diabetes mellitus (DM) to improve blood sugar control adjunct to diet and exercise.
Current international guidelines as the ADA guidelines 2025 recommend SGLT2i for people with T2D and established ASCVD or indicators of high ASCVD risk, HF, or CKD, independently of A1C values and metformin use, and in consideration of person-specific factors”.
- Following the Introduction, I suggest including a paragraph discussing the ongoing debate regarding the expression of SGLT2i at the cardiac level (in cardiomyocytes), highlighting both supporting and opposing evidence (opposing: PMID: 32613148, 20045149; supporting: PMID: 30291295, 36096423, 38247224).
Re: We welcomed the referee's suggestion, and an extensive paragraph on this topic has been added to the review. Please, see paragraph: 2. Effects of SGLT2i on cardiovascular outcomes
“
The beneficial effect that these molecules would have at the cardiac level could depend on several factors. Still suggestive and interesting is the hypothesis that these drugs may directly inhibit SGLT2, which is also present in the myocardium: the expression of the genes coding for this cotransporter has not yet been proven and conflicting experiences exist in the literature.
In the study by Marfella et al. which evaluated the expression of SGLT2 in human cells from diabetic and non-diabetic patients and in an AC16 myocardiocyte line from patients undergoing heart transplantation using immunohistochemistry, immunofluorescence and SGLT2 quantization with both real-time reverse transcription-polymerase chain reaction and Western blot analysis showed the presence of SGLT2 in patients with end-stage decompensation and that this expression is overexpressed in patients with diabetes mellitus; additionally when cardiomyocytes were submitted to high concentrations of glucose, this leads to overexpression of SGLT2 in these cells (11).
This overexpression was also demonstrated in the study by Scisciola et al. In this case, overexpression of the SGLT2 gene and protein was marked in patients with low-flow, low gradient aortic stenosis (LF-LG AS), and this overexpression appeared to be related to important changes in cardiac metabolism: in particular, there was less utilisation of fatty acids and their oxidation to produce energy, with greater utilisation of glucose as an energy substrate and subsequent accumulation of lipids in the myocardium. This, in turn, led to increased expression of SGLT2 which led to a greater metabolic shift towards glucose utilisation with reduced ATP production and functional failure of the myocardium itself. This naturally led to a progressive deterioration of cardiac function and failure. (12)
These results were finally confirmed in a study on cardiomyocytes derived from human induced pluripotent stem cells (hiPSCs) that were given high doses of glucose to induce hypertrophy and were subsequently treated with empagliflozin: during the first phase of the study, there was increased expression of SGLT1, SGLT2 and NPPB, which also augmented with increasing cell size in immunofluorescence images, and this led to alterations in intracellular calcium concentrations that resulted in reduced sensitivity of myocardial fibres to this ion. Starting of empagliflozin therapy showed a reduction in cytosolic calcium levels and a normalisation of cardiomyocyte contractility, supporting the thesis of a direct effect of this molecule on the myocardiocyte independent of other metabolic activities. (13)
In contrast, in a study that isolated atrial and ventricular myocardiocytes from 88 patients undergoing cardiac surgery and when these cells were submittet to an insulin-containing solution by assessing mRNA levels, no expression of SGLT2 was demonstrated, but only of SGLT1, GLUT1 and GLUT4. (14).
This has led several scholars to believe that the effect of these molecules is not due to their direct action on specific receptors present on the myocardium but to other side effects such as natriuresis/diuresis, improved cardiac energy metabolism due to the production of ketone bodies, inhibition of the sympathetic nervous system, inhibition of the Na/H-exchanger, improvement of hyperuricaemia, inhibition of SGLT1 decreasing epicardial fat mass, increasing erythropoietin levels, increasing circulating pro-vascular progenitor cells, decreasing oxidative stress, and improving vascular function. (15)”
- Given the growing body of evidence in this field, it may be useful to split the section on cardiovascular outcomes into two distinct parts: “Effects of SGLT2i on Acute and Chronic Heart Failure” and “Effects of SGLT2i on Acute Myocardial Infarction.” Both sections should be expanded and updated with recent references (PMID: 39262451, 36036746).
Re: We thank the Reviewer for this suggestion. The paragraphs have been separated accordingly in “SGLT2i and Acute and Chronic Heart Failure” and “SGLT2i and Acute Myocardial Infarction”, and literature has been updated by including the suggested references.
- In the section “Effects of SGLT2i on Renal Outcomes,” consider adding a few sentences about the beneficial effects of SGLT2i in preventing contrast-induced acute kidney injury (CI-AKI), both in patients with chronic coronary syndrome and in those presenting with acute myocardial infarction.
Re: This paragraph has been modified by adding the following sub-section “ 2.3. SGLT2i Contrast Induced Acute Kidney Disease”.
- Please include a new section dedicated to the most recent and emerging evidence on the benefits of SGLT2i in patients with aortic stenosis undergoing transcatheter aortic valve implantation (TAVI), with and without diabetes (PMID: 39574095, 40162639, 38247224, 39985508).
Re: Also, a new and updated paragfraph “2.3. SGLT2i and Aortic Stenosis” has been added to the revised review
- Kindly revise Figure 1 to reflect these suggested changes.
Re: Figure 1 has been revised to include the suggested changes. Moreover, a new figure has been added. Please see revised figures 1 and 2

Reviewer 3 Report
Comments and Suggestions for Authors
This review is well -written and provides a comprehensive summary of the current therapeutic effects of SGLT2 inhibitors across various clinical settings beyond glucose lowing in diabetes. However, one concern remains: an increasing number of studies suggest that SGLT2 may also be expressed in other cell types, such as glomerular mesangial cells and podocytes, although it is typically localized to proximal tubules. It would be beneficial to mention this possibility or provide comments on these findings in the review.
Author Response
This review is well -written and provides a comprehensive summary of the current therapeutic effects of SGLT2 inhibitors across various clinical settings beyond glucose lowing in diabetes.
Re: We would first like to thank the Referee for the appreciation of our work.
However, one concern remains: an increasing number of studies suggest that SGLT2 may also be expressed in other cell types, such as glomerular mesangial cells and podocytes, although it is typically localized to proximal tubules. It would be beneficial to mention this possibility or provide comments on these findings in the review.
Re: we thank the Reviewer for this suggestion. Accordingly, the paragraph on the effects of SGLT2i on renal function has been modified to include a comment on this topic, as specified below: "
Furthermore, as is well known, mesangial cells and pericytes play a key role in regulating capillary blood flow and the passage of substances from the glomerular membrane. Several studies have reported the presence of SGLT2 in bovine retinal pericytes and mesangial cells (49). Similarly, SGLT2 has also been shown to be present in renal proximal tubule cells, mesangial cells and pericytes of non-diabetic mice (50); and this could lead to a metabolic perturbation promoted by the mTORC1-signalling pathway that appears to be mitigated by SGLT2 inhibition (51)
In addition, the study by Li et al. showed how SGLT2 inhibition leads to increased podocyte protection in diabetes mellitus subjects via mitochondrial membrane-mediated energy balancing.(52)"
Round 2
Reviewer 1 Report
Comments and Suggestions for Authors
Usefull paper, well written, very good review, great job.
Just add a table resuming results in CV protection and kidney disease ?
Author Response
We would like to thank the Referee for his kind comment and suggestion.
"Usefull paper, well written, very good review, great job.
Just add a table resuming results in CV protection and kidney disease ?"
We have added a new table to summarise the results on the achievement of the target for the main cardiovascular risk factors and the results on renal protection related to the use of SGLT2i
I hope you enjoy it
Kind regards
Reviewer 2 Report
Comments and Suggestions for Authors
The authors have made a substantial effort to address the reviewers' comments. The manuscript has noticeably improved. I have no further comments.
Author Response
The authors have made a substantial effort to address the reviewers' comments. The manuscript has noticeably improved. I have no further comments.
We would like to thank you for your kind comments and suggestions, we are happy that you like the final work